# Chiral Separation of Apremilast by Capillary Electrophoresis Using Succinyl-β-Cyclodextrin—Reversal of Enantiomer Elution Order by Cationic Capillary Coating

**DOI:** 10.3390/molecules28083310

**Published:** 2023-04-08

**Authors:** Zoltán-István Szabó, Beáta-Mária Benkő, Ágnes Bartalis-Fábián, Róbert Iványi, Erzsébet Varga, Levente Szőcs, Gergő Tóth

**Affiliations:** 1Faculy of Pharmacy, George Emil Palade University of Medicine, Pharmacy, Science, and Technology of Targu Mures, Gh. Marinescu 38, 540139 Târgu Mureș, Romania; 2Sz-Imfidum Ltd., nr. 504, 525401 Lunga, Romania; 3University Pharmacy Department of Pharmaceutical Administration, Semmelweis University, Hőgyes E. 9, H-1085 Budapest, Hungary; 4Cyclolab Ltd., Illatos út 7, H-1097 Budapest, Hungary; 5Department of Pharmaceutical Chemistry, Semmelweis University, Hőgyes E. 9, H-1085 Budapest, Hungary

**Keywords:** apremilast, reversal of enantiomeric elution order, cyclodextrin, CE, capillary coating

## Abstract

A stereospecific capillary electrophoresis method was developed for the separation of the novel, antipsoriatic agent, apremilast (APR). Six anionic cyclodextrin (CD) derivatives were screened for their ability to discriminate between the uncharged enantiomers. Only succinyl-β-CD (Succ-β-CD) presented chiral interactions; however, the enantiomer migration order (EMO) was unfavorable, and the eutomer, *S*-APR, migrated faster. Despite the optimization of all possible parameters (pH, cyclodextrin concentration, temperature, and degree of substitution of CD), the method was unsuccessful for purity control due to the low resolution and the unfavorable enantiomer migration order. Changing the direction of electroosmotic flow (EOF) by the dynamic coating of the inner surface of the capillary with poly(diallyldimethylammonium) chloride or polybrene resulted in EMO reversal, and the developed method could be applied for the determination of *R*-APR as the enantiomeric purity. Thus, the application of the dynamic capillary coating offers a general opportunity for enantiomeric migration order reversal in particular cases when the chiral selector is a weak acid.

## 1. Introduction

Apremilast (APR; Figure 1) is an orally active, novel drug used for the treatment of psoriasis and psoriatic arthritis. It acts as a selective phosphodiesterase 4 (PDE4) inhibitor, downregulating the characteristic inflammatory response in the psoriasis [1,2]. The oral small-molecule PDE4 inhibitor blocks the degradation of cyclic adenosine 3′,5′-monophosphate (cAMP), and with the increasing level of intracellular cAMP in PDE4-expressing cells, it modulates the system of pro-inflammatory (e.g., reduced TNF-alfa or IL-23 expression) and anti-inflammatory (e.g., increased IL-10 expression) mediators [3,4]. Structurally, APR may be regarded as a thalidomide-related compound, as it shares the phthalimide ring and presents a single chiral carbon atom [5], but in contrast to thalidomide, the glutarimide ring is absent. This makes the molecules more stable against racemization in vivo [6]; thus, the enantiopure form was approved by the Food and Drug Administration (FDA) in 2014 and European Medicines Agency (EMA) in 2015 for the treatment of psoriasis and psoriatic arthritis. For this drug, the more potent *S*-enantiomer is marketed; therefore, the determination of the *R*-antipode is a requirement [7,8]. Although several chromatographic methods are described or at least mentioned for the achiral [9,10] and chiral separation of apremilast [11,12,13,14,15,16], based on the authors’ knowledge, enantioseparation of APR with capillary electrophoresis was not yet described.

Although LC is the most common choice for enantioseparations, alternative techniques such as capillary electrophoresis (CE) present several advantages over LC. Chiral CE determinations are greener and cheaper compared to HPLC analyses, with a minimal sample requirement, very low reagent and solvent consumption, fast analysis times, and generally higher efficiency. There is also no need for costly chiral columns, as in most cases, a chiral selector is simply dissolved in the background electrolyte (BGE) [17]. Due to the low volume of BGE required, even a small amount of the chiral selector is sufficient for CE measurements [18]. Several chiral selectors are available in CE, such as cyclodextrins (CDs), macrocyclic antibiotics, proteins, and crown ethers. Among these, CD derivatives are the most efficient and commonly used chiral selectors in CE [19]. The wide variety of structurally diverse (neutral and charged) derivatives, low UV cut-off, and relatively low price make them indispensable for chiral CE applications [20,21].

One often overlooked aspect of chiral separations is enantiomer migration/elution order, which is especially important since the main component can disturb the quantification of the minor component(s) (the impurity) via system overloading. The ideal elution/migration order is that the distomer is followed by the eutomer. In earlier days, it was thought that in LC, enantiomer elution order reversal is only possible, when a chiral selector is changed with its counterpart in stereochemical configuration [22]. Research and developments in recent years, however, have shown that for several chiral selectors, a change in the mobile phase composition can also alter the enantioselectivity of the chiral selector, which can often result in enantiomer elution order reversal [23,24,25,26]. However, this approach still works on a trial-and-error basis, and prediction of enantiomeric elution order is still very difficult, if not impossible.

Compared to chromatographic techniques, chiral CE offers an additional, inherent alternative for enantiomer migration order (EMO) reversal. Because of the vectorial property of the effective mobility, EMO reversal can be achieved by simply reverting the polarity of the high-voltage power supply or changing the direction of the electroosmotic flow (EOF), provided that the analytes are still detectable [22,27,28].

The direction and magnitude of EOF can be modified by using capillary coatings, which are generally categorized as dynamic, semi-permanent, and permanent, based on the attachment of the coating molecules to the capillary surface. In the case of dynamic coatings, the coating agent is part of the BGE, while semi-permanent coatings are physically absorbed to the inner surface of the capillary wall with an initial flush of a coating solution and are usually stable for multiple runs, without regeneration. Permanent coatings are based on the irreversible, covalent bonding between the coating molecules and the capillary wall [29,30].

Compared to covalently modifying the capillary inner surface, dynamic coatings, are easier to prepare by simple rinsing steps and can be successfully applied for eliminating, reversing or finetuning the electroosmotic flow, as well as diminishing protein adsorption to the capillary [30]. Compounds often used for dynamic coating are positively charged additives (amines, cationic surfactants, cationic polymers, and ionic liquids), which readily interact with the negatively charged surface silanol groups, changing capillary surface charge and resulting in a reversed EOF. Another possibility is to use hydrophilic polymers (polyethylene oxide, cellulose-derivatives, and polyvinyl alcohol), which diminish or eliminate the EOF [31].

The present work reports on the first CE enantioseparation of APR using CDs as chiral selectors. Since the only investigated CD derivative that exhibited chiral interactions with the enantiomers yielding low selectivity and unfavorable migration order, dynamic capillary coating techniques were employed to achieve EMO reversal. This specific case of enantioseparation highlights the advantages of CE over other chiral separation techniques due to the ease of EMO reversal.

## 2. Results and Discussion

Since APR is a neutral compound under typical CE conditions, only charged CD derivatives can be used to provide the necessary velocity difference for the enantioseparation. During the preliminary experimental runs, 10 widely available anionic CDs (CM-α-CD DS = 3.5, CM-β-CD DS = 3.5, SBE-β-CD DS = 4, SBE-β-CD DS = 6.3, SBE-β-CD DS = 10, Succ-β-CD DS = 3.5, SP-β-CD DS = 4, and S-β-CD DS = 9) were applied as potential chiral selectors at five different concentration levels at pH = 7.0, using 50 mM phosphate buffer and applying 20 kV voltage, while maintaining the capillary temperature at 25 °C.

Out of all the anionic CDs used in this study, only Succ-β-CD displayed chiral recognition towards the APR enantiomers. This is interesting, since under similar conditions, structurally related compounds, such as thalidomide [32], pomalidomide [33], and lenalidomide [34], were baseline-separated using SBE-β-CD, and CM-β-CD (apart from lenalidomide). Succ-β-CD was not applied in any of the abovementioned cases.

In case of APR, the unfavorable, eutomer first EMO, and the low *R*_s_ values obtained (*R*_s_ = 1.45 at a 20 mM Succ-β-CD concentration in 50 mM phosphate buffer pH = 7.0) made the determination of low concentrations of the chiral impurity impossible. Higher concentrations of the chiral selector led only to a small increase in *R*_s_ value (*R*_s_ = 1.73 at 30 mM Succ-β-CD) and almost doubled the migration times of the enantiomers. Electropherograms obtained with increasing concentrations of Succ-β-CD (DS = 3.5) are depicted in Figure 2.

All the applied anionic CDs are multicomponent mixtures, with random substitutions. Only the average DS value or the DS range is declared in these cases. Differences in the DS value of the same derivatized CD can have a big impact on the enantioseparation performance of the given chiral selector [27,35,36,37] and can even lead to several problems, when attempting to repeat earlier developed methods, due to the batch-to-batch variations of the chiral selector [38]. Kim et al. compared the enantioseparation performances of single isomeric mono-, di-, and tri-Succ-β-CD for the enantioseparation of catechin in high pH buffers, where the analyte is mostly in its anionic form. The authors noted several differences between the performances and the application ranges of these chiral selectors [39]. Thus, to further increase enantioresolution, custom-synthesized Succ-β-CD with two different DSs (DS = 5 and DS = 10) were also tried as chiral selectors. In our case, applying these chiral selectors with higher DS, using the same CE conditions led only to increased migration times, without chiral separation (Figure 3). A similar trend was also observed in our earlier publication for the enantioseparation of the structurally similar lenalidomide [34], where an increase in declared average DS for SBE-β-CD led to a decrease in enantioseparation performance. The two works also draw attention to the importance of DS in chiral CE. Furthermore, optimizing DS can be an important parameter to consider.

It was observed that increasing the concentration of the BGE resulted in only a minor improvement in enantiomeric resolution. However, it also led to an increase in migration time and generated higher current. This was due to the generated currents exceeding 120 μA, resulting in baseline instabilities.

Since none of the abovementioned approaches led to a significant increase in enantioresolution and the EMO was also unfavorable, attempts were made to reverse it. One of the main advantages of CE in enantioseparations, when compared to chiral LC, is that EMO reversal is possible even without manipulating the affinity of the chiral selector towards the individual enantiomers or by exploiting the vectorial property of the mobility difference between the enantiomers. The theoretical background and applicability of this approach has been described in detail by Chankvetadze et al. in earlier publications [22,40,41,42]. In some of our recent publications, we also exploited the possibility of EMO reversal in the case of chiral separation of neutral and cationic enantiomers with anionic CDs. In these publications, EOF suppression was obtained by employing low pH acidic BGEs to suppress the dissociation of silanol groups of the inner of the untreated silica capillary. Due to the suppressed EOF, reversing the polarity of the separation system resulted in EMO reversal [27,28]. However, in the present case, this simple, straightforward approach was not applicable, since Succ-β-CD is practically uncharged at the low pH values (pH < 2.5) needed for EOF suppression; thus, under these conditions, it cannot be used for the chiral separation of the neutral APR enantiomers.

Another way to achieve EMO reversal through modulating EOF is by employing capillary coatings. Essentially, the inner wall of the silica capillary can be coated either by physical adsorption or by covalent bonding. Out of these two approaches, the former excels by its simplicity and ease of use, which in most cases consists of rinsing a coating solution through the capillary to cover the capillary wall. Because the coating agent is attached to the capillary wall through adsorption, in most cases the coating needs to be regenerated after several electrophoretic runs or the surface concentration needs to be maintained by adding the coating agent to the BGE.

The most often used dynamic capillary coatings include hexadimethrine bromide (Polybrene, PB; Figure 4A), polydiallyldimethyl-ammonium chloride (PDADMAC; Figure 4B), cellulose-derivatives, and polyethylene oxide (PEO) [30]. Ready-to-use kits to perform dynamic capillary coating are also commercially available, and stabilization of capillary coatings using successive multiple ionic layers-type coatings are also widely described [43]. In the present study, simple, single-layer capillary coatings were employed, to achieve EMO reversal by reversing the direction of the EOF and switching the polarity of the CE system.

In the bare fused-silica capillary, without the chiral selector, the neutral APR enantiomers are carried to the detector by the cathodic EOF. Upon the addition of Succ-β-CD to the BGE, enantiodiscrimination is observed, and upon complexation with the anionic CD, both enantiomers are detected after the EOF. In this case, the enantiomeric impurity of *R*-APR migrates lasts, because upon complexation with the chiral selector it has a higher anodic mobility than the eutomer, *S*-APR. Dynamically coating the capillary with cationic polymers results in an anodic EOF; thus, by polarity switching, the EMO is also reversed, with *R*-APR being the first-migrating enantiomer, because of its higher anodic mobility.

Using PDADMAC and PB to generate cationic capillary coatings, it was observed that after an initial flush with the polymeric solution and stabilization, preconditioning with the cationic polymer solution was also needed after each injection, otherwise, the reverse EOF diminished or disappeared. Upon application of a PDADMAC coating, as expected, the EMO was reversed; however, the coating lacked long-term robustness and stability, and a migration time drift was observed with subsequent injections, even after reapplying the coating upon preconditioning. Conversely, we did not encounter similar issues with the PB coating.

Subsequently, we carried out an optimization of the injection volume and the temperature. The effect of the temperature on chiral separation was tested at three different values (20, 30, and 40 °C). The results are summarized in Table 1.

Higher temperatures resulted in shorter migration times, but at the expense of the decreased resolution and the increased current (around 75 μA at 20 °C vs. around 92 μA at 40 °C). The latter could lead to baseline instabilities and detachment of the capillary coating. The best resolution value was observed at 20 °C, and this temperature was considered optimal.

During the injection optimization process, we maintained a constant pressure of 50 mbar and varied the duration time between 1, 2, and 4 s. Among the injection volumes tested, we found that the combination of 50 mbar and 4 s produced the best results.

Using the optimized method conditions (25 mM Succ-β-CD (DS = 3.5) in 50 mM phosphate buffer with pH 7.0; dynamic capillary coating—PB, which required renewal before each injection; a −18 kV applied voltage; a 20 °C capillary temperature; hydrodynamic injection at 50 mbar × 4 s), a high resolution (*R*_s_ > 3) was achieved within 12 min. Using these conditions, we were able to quantify the 0.10% *R*-APR enantiomer impurity in an *S*-APR sample (Figure 5).

The developed method’s analytical performance was evaluated according to the current guidelines. The limit of detection (LOD) and the limit of quantification (LOQ) values for *R*-APR were determined at concentration values yielding a signal-to-noise ratio of 3:1, and 10:1, respectively. The LOD was 1.0 µg/mL (corresponding to 0.05% of impurity in a 2000 µg/mL *S*-APR sample), while the LOQ was 2.0 µg/mL (0.10% of impurity).

To evaluate the repeatability of migration times and peak areas, samples were prepared at 2.0 μg/mL (0.1%) *R*-APR, in the presence of the eutomer at a sample concentration (2000 μg/mL). The repeatability of the method, calculated as an RSD for migration times were 1.00% and 0.84% for *R*- and *S*-APR, respectively. In the case of the peak area, the RSD for *R*-APR was 1.96% (*n* = 6). Inter-day precision values for migration times were RSDs of 1.09% and 0.86% for *R*- and *S*-APR, respectively, while it was 2.17% for the peak area of the distomer (*n* = 12). Repeating the analysis on a different capillary prepared from the same batch revealed RSD values of 1.32% and 1.19% for *R*- and *S*-APR, respectively, for migration times and 2.23% for the peak area of *R*-APR. Overall, these results demonstrate the method’s excellent analytical performance, with high sensitivity, repeatability, and precision.

## 3. Materials and Methods

### 3.1. Materials

*R*- and *S*-APR were purchased from Beijing Mesochem Technology Co., Ltd. (Beijing, China). Acetonitrile (ACN) of gradient grade was purchased from Merck (Darmstadt, Germany). Acetic acid, Tris, NaOH, NaCl, HCl, phosphoric acid, disodium hydrogen phosphate, and monosodium hydrogen phosphate (Sigma-Aldrich, Budapest, Hungary) used for the preparation of buffer solutions and rinsing solutions were of analytical grade. All reagents were used without further purification. Bidistilled Millipore water was used throughout this study.

Sulfated-β-CD with a degree of substitution (DS) of ~11 was purchased from Sigma-Aldrich (Budapest, Hungary).

All other CD derivatives with various DS values: carboxymethylated-α-CD (CM-α-CD; DS: ~3.5), CM-β-CD (DS: ~4), sulfopropylated-β-CD (SP-β-CD; DS: ∼4), sulfobutyl-ether-β-CD (SBE-β-CD; DS: ~4, ~6.3, and ~10), as well as succinyl-β-CD (Succ-β-CD), were products of Cyclolab (Budapest, Hungary).

Commercially available Succ-β-CD had a declared average DS of ~3.5. To synthesize Succ-β-CD with custom DS values (DS of ~5 and DS of ~10), the following procedure was employed: native β-CD was dissolved in pyridine at room temperature. The desired DS was achieved by adding succinic anhydride at various mole ratios to obtain DS values of ~5 and ~10, 6 and 12 equivalents of succinic anhydride should be added to the rection mixture, respectively. The reaction was followed with thin layer chromatography (eluent ratio of dioxane:ammonia = 1:1; *R*_f_ of β-CD was around 0.5. The lower the *R*_f_ for Succ-β-CD, the higher the DS of the synthesized product). As far as we know, randomly substituted Succ-β-CDs with higher DS have not yet been used as chiral selectors.

After overnight stirring, pyridine was evaporated, and methanol was added to obtain a solution. After adding acetone, the product crystallized from the solution. The crystals were filtered through a G3 filter and were washed with acetonitrile.

The determination of the DS and the residual succinic anhydride content of the obtained product were performed with CE, using the following parameters and conditions: A fused silica capillary with a total length of 33.5 cm (effective length: 25.0 cm), an internal diameter 50 µm, and an O.D. of 375 µm (Composite Metal Services Ltd., Worcestershire, UK). Tris (60 mM) and para-toluene sulfonic acid (30 mM; pH: ~8.2) were used as the BGE. Hydrodynamic injections were performed at 50 mbar × 4 s. The temperature of the capillary cassette was maintained at 25 °C, and 20 kV voltage was applied. Indirect UV detection was performed at 350 nm (bandwidth: 20 nm), using a reference wavelength at 225 nm (bandwidth: 20 nm).

### 3.2. Enantioselective CE Conditions

CE experiments were performed on an Agilent G1600 CE instrument (Agilent Technologies, Waldbronn, Germany), equipped with a photodiode array detector, and ChemStation 7.01 was the software used for data handling. An untreated fused-silica capillary (50 µm id, a 48.5 cm total length, and a 40 cm effective length) from Agilent (Waldbronn, Germany) was used in the initial experiments. The dynamic coating techniques are described in the following section. Conditioning of new capillaries was conducted by flushing with 1 M NaOH for 30 min followed by 0.1 M NaOH for 30 min and water for 15 min. In the scouting phase before all runs, the capillary was preconditioned by flushing with 0.1 M NaOH (3 min), water (2 min), and BGE (5 min). UV detection was performed at 210, 230, and 280 nm. Unless otherwise stated, injections were carried out by applying a pressure of 50 mbar for 2 s.

### 3.3. Dynamic Coating Techniques

#### 3.3.1. Polydiallyldimethylammonium Chloride (PDADMAC) Monolayer Coating

The coating procedure using PDADMAC was applied based on the work of Kpaibe et al. with some modifications [44]. First, a 0.2% (*w*/*w*) PDADMAC solution at pH = 8.3 with a 1.5 M ionic strength was prepared using TRIS, NaCl, and HCl. The solution was stored at 4 °C for a maximum of 10 days. Before use, it was filtered through a 0.45 μm-pore-size Nylon syringe filter.

Before coating, the new capillary was rinsed with NaOH 1 M for 30 min, NaOH 0.1 M for 30 min, and water for 10 min. Coating of the capillary was performed by flushing a 0.2% (*w*/*w*) PDADMAC solution for 120 min. The coating was stabilized by applying −10 kV for 10 min. Eventually, the capillary was rinsed with water for 10 min and BGE for 5 min.

Between analyses, the capillary was preconditioned with a 0.02% (*w*/*w*) PDADMAC solution for 1 min, water for 1 min, and BGE for 3 min.

#### 3.3.2. Polybrene (PB; Known as Hexadimethrine Bromide) Monolayer Coating

The coating procedure with PB was applied based on the previous works of Haselberg et al. [45,46] with some modifications. Briefly, a 10 *w*/*v%* PB stock solution was prepared in water and then filtered through a 0.45 μm-pore-size Nylon syringe filter.

Capillaries were coated by subsequently rinsing with a 10% (*w*/*v*) PB solution for 10 min, and after 5 min of wait time, the capillary was rinsed with water for 5 min. The capillary was then ready for the CE analysis. Between runs, coated capillaries were flushed with a 0.1% (*w*/*v*) PB solution, water, and BGE for 3 min each before the injection.

### 3.4. Preparation of BGE and Sample Solutions

Preliminary experiments were performed in a 50 mM phosphate buffer adjusted to pH of 7 with NaOH (0.1 M); the BGE contained CDs in various concentrations (0, 3, 6, 15, 20, and 25 mM), the applied voltage was 20 kV, and the capillary was maintained at 25 °C.

Stock solutions of 10 mg/mL *R*- and *S*-APR were prepared using acetonitrile as the solvents. During the preliminary studies, it was revealed that the best peak shapes were obtained if the sample solution contained the least amount of the organic solvent. If the sample solution contained higher concentrations of the organic solvent, peak distortion or even peak splitting was observed. Thus, our approach was to dissolve the enantiomers in a suitable solvent to obtain a higher-concentration stock solution, and upon further dilution, try to minimize the solvent used. ACN was chosen because it allowed us to prepare 10 mg/mL initial stock solutions. The solubility of the enantiomers was lower in alcohols. Sample solutions were prepared by diluting the stock solution with water. In the preliminary experiments, an enantiomeric ratio of *R*:*S*-APR of around 1:3 was prepared for the analysis of the EMO. During the CD screening phase, 0.01% DMSO (*v*/*v*) was used as the EOF marker. After selecting Succ-β-CD as the optimal chiral selector, the EOF marker was no longer used in the sample.

## 4. Conclusions

Obtaining the ideal EMO is a critical challenge when aiming to determine enantiomeric purity of chiral drugs. Frequently used chiral selectors in chromatography do not allow for deliberate reversal of the enantiomer elution sequence. In contrast, in CE, the EMO can be deliberately altered, which facilitates the method development. During the development of an enantioselective CE method for APR, enantiomeric recognition was observed only with Succ-β-CD as a chiral selector, and even in this case, only low *R_s_* values were obtained with a non-ideal EMO. The latter hindered the determination of low concentrations of the distomer (*R*-APR), and even the application of custom-synthesized Succ-β-CDs, with higher DSs, were not able to increase enantioresolution. Since Succ-β-CD is a weak acid, the EMO cannot by reversed by EOF suppression at low-pH BGE, because the selector becomes uncharged. Thus, EOF needed to be altered by modifying the silica capillary wall. While testing PDADMAC- and PB-based dynamic coatings, only the PB coating proved to be suitable for routine application of EMO reversal and was successfully applied for the determination of low concentration of the distomer. This approach further underlines the ease and flexibility of the enantioselective CE method development and underlines its importance in chiral analysis.

## Figures and Tables

**Figure 1 molecules-28-03310-f001:**
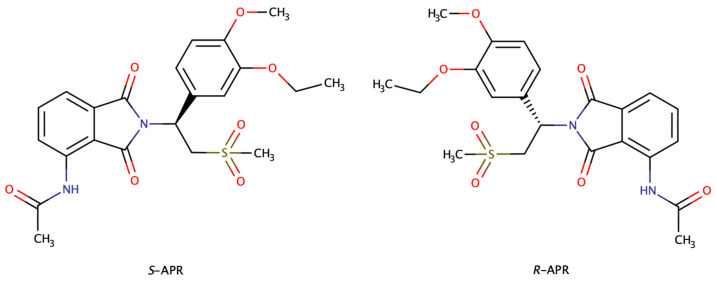
Constitutional formulas of APR enantiomers.

**Figure 2 molecules-28-03310-f002:**
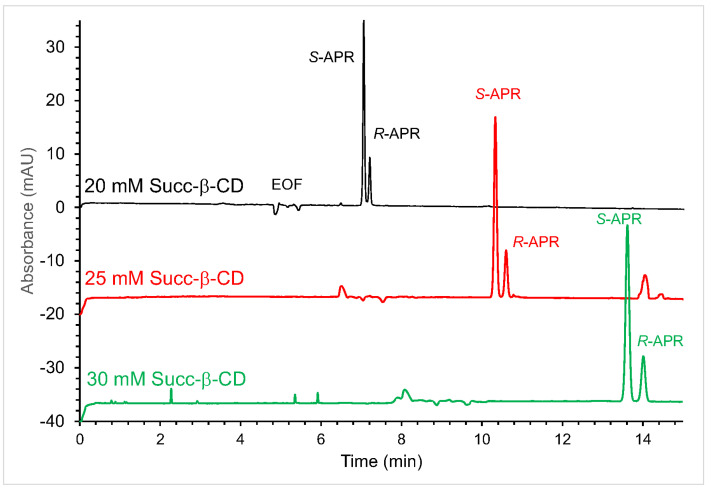
Enantioseparations of APR with Succ-β-CD (DS = 3.5) at 20 mM, 25 mM, and 30 mM. (CE conditions: 50 mM phosphate buffer (pH 7.0), supplemented with the indicated concentration of Succ-β-CD; capillary temperature: 25 °C; hydrodynamic injection: 50 mbar × 2 s; detection: 210 nm and 20 kV).

**Figure 3 molecules-28-03310-f003:**
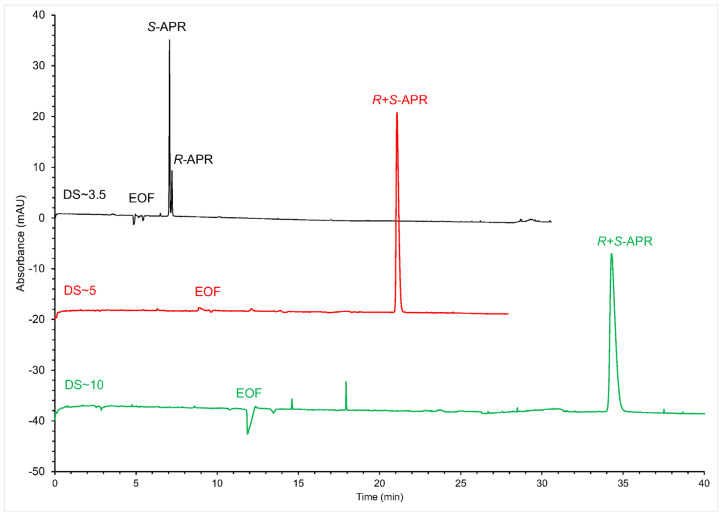
Enantioseparations of APR applying Succ-β-CD with different DS values. (CE conditions: 50 mM phosphate buffer pH 7.0, supplemented with 20 mM Succ-β-CD with the indicated DS; capillary temperature: 25 °C; hydrodynamic injection: 50 mbar × 2 s; detection: 210 nm and 20 kV).

**Figure 4 molecules-28-03310-f004:**
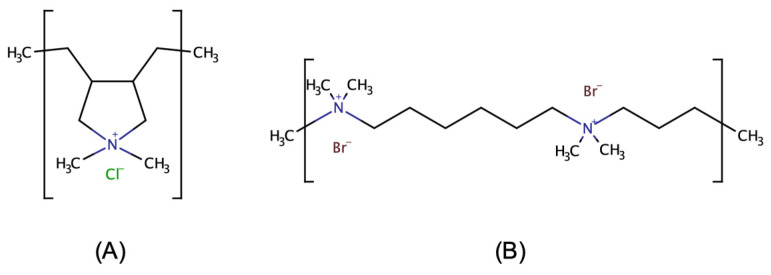
The chemical structures of the applied cationic polymers as coating agents: (**A**) PDADMAC; (**B**) PB.

**Figure 5 molecules-28-03310-f005:**
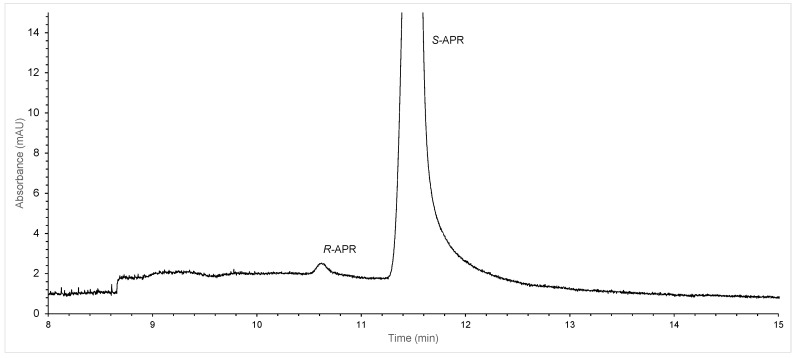
Electropherogram obtained using the optimized parameters for the determination of 0.1% *R*-APR in an *S*-APR sample. CE conditions: PB-coated capillary; BGE: 50 mM phosphate buffer (pH 7.0), supplemented with 25 mM Succ-β-CD with the indicated DS; capillary temperature: 20 °C; hydrodynamic injection: 50 mbar × 4 s; detection: 230 nm and −18 kV (reverse polarity). Sample concentrations: 2 µg/mL *R*-APR and 2000 µg/mL *S*-APR.

**Table 1 molecules-28-03310-t001:** Effects of the temperature on migration times of the analytes and enantioresolution.

Temperature	t*_R_*_-APR_ (min)	t*_S_*_-APR_ (min)	*R_s_*
20 °C	10.92	11.80	4.35
30 °C	10.30	11.06	3.72
40 °C	9.19	9.75	2.76

t*_R_*_-APR_—migration time of *R*-APR; t*_S_*_-APR_—migration time of *S*-APR; *R*_s_—resolution between enantiomers.

## Data Availability

All relevant data are included in the article.

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
