# Peer review of "Chiral Separation of Apremilast by Capillary Electrophoresis Using Succinyl-β-Cyclodextrin—Reversal of Enantiomer Elution Order by Cationic Capillary Coating"

_molecules, 2023, doi:10.3390/molecules28083310_

Round 1

Reviewer 1 Report

This manuscript provides a new method for enantioseparation of apremilast using chiral CE. The manuscript is very useful to the readers given such a CE method did not exist before. The following are my comments which should be addressed is the manuscript.

1. The authors have shown reversal of elution order which is extremely important in chiral chromatography. However since in this case it is known which one is the active enantiomer and as apremilast has been separated using chiral LC before, can the authors highlight why this would be of interest? Also if this method in any way actually betters the analysis of aprelimast compared to previous analytical methods?

2. The authors have conducted a temperature study and selected 20 C as their optimized temperature. It is commonly known that lower temperatures increase resolution in chiral chromatography and CE at the expense of increased retention times. However since at all 3 temperatures tested the components are baseline resolved why did the authors not chose the higher temperature as it reduces run time. 

3. The authors should conducted a study to figure out the limit of detection for the method and compare it to the chiral LC method available.

Author Response

This manuscript provides a new method for enantioseparation of apremilast using chiral CE. The manuscript is very useful to the readers given such a CE method did not exist before. The following are my comments which should be addressed is the manuscript.

1. The authors have shown reversal of elution order which is extremely important in chiral chromatography. However since in this case it is known which one is the active enantiomer and as apremilast has been separated using chiral LC before, can the authors highlight why this would be of interest? Also if this method in any way actually betters the analysis of aprelimast compared to previous analytical methods?

Although the manuscript describes the first enantioseparation method of apremilast using capillary electrophoresis, which is a much cheaper alternative to chromatography-based separation methods, the real value of the manuscript lies in underlying that in the case of CE, enantiomer migration order can be reversed universally. No other separation technique offers this advantage in chiral separation.

The manuscript thus exemplifies a case, where, even if all approaches to increase resolution or alter the selectivity of an enantioseparation method fail; due to the inherent characteristics of CE (electrophoretic mobility being a vectorial property), enantiomer migration order reversal is still attainable.

2. The authors have conducted a temperature study and selected 20 C as their optimized temperature. It is commonly known that lower temperatures increase resolution in chiral chromatography and CE at the expense of increased retention times. However since at all 3 temperatures tested the components are baseline resolved why did the authors not chose the higher temperature as it reduces run time. 

As the reviewer noted, at higher temperatures a reduction of analysis time was observed. However this was attained at the expense of increased current (around 75 μA at 20 °C vs around 92 μA at 40 °C), which could have led to baseline instabilities and the detachment of capillary coating.

This is now also included in the revised manuscript.

3. The authors should conducted a study to figure out the limit of detection for the method and compare it to the chiral LC method available.

Based on the comments of the reviewers, the LOD and LOQ of the method were checked, and determined to be 1.0 μg/mL (corresponding to 0.05%), and 2.0 μg/mL (corresponding to 0.1%), respectively. As expected, the values obtained are about twice as high as the ones obtained using our earlier described LC method.

This is now also included in the revised manuscript.

Reviewer 2 Report

The manuscript describes the separation of apremilast by CE using CDs as additives. The paper is clearly written, and the results are good, in that the authors managed to develop a CE BGE system suitable for the enantiomeric separation of apremilast. However, some validation tests are needed.

What is the typical purity limit of the R-APR in commercial S-APR samples? Is the typical limit 0.15% or even lower? Please check the LOD of R-APR in the presence of required amounts of S-APR. In the figure caption of Figure 3, the R-APR concentration is said to be 0.1% (not 0.15% as in the text). Please give repeatability and reproducibility data (on a different capillary) for the 0.1% enantiomeric purity analysis. Please also make a robustness test.

The Chinese patent CN107628983A https://patents.google.com/patent/CN107628983A/en seems to have a procedure for making apremilast with an enantiomer purity of more than 99.9%. Please check.

Please give some references to the lack of interconversion of the S-form to the L-form (this is something industrial people will be interested in). See eg. https://www.ema.europa.eu/en/documents/assessment-report/otezla-epar-public-assessment-report_en.pdf The assessment of the chiral purity in that report from 2014 was claimed to be done by HPLC.

-Why were the samples diluted in acetonitrile? How was the type of solvent optimized? Typically, sample stock solutions for CE are done in methanol or ethanol.

-What was used as the EOF marker in the work? The EOF peaks visible in Figure 2 are probably due to ACN present in the sample. Please mark the EOF with an arrow in the electropherograms. The same holds for all the other electropherograms.

-Please check these recent papers, related to the topic:

-A review paper: “New Trends in the Quality Control of Enantiomeric Drugs: Quality by Design-Compliant Development of Chiral Capillary Electrophoresis Methods” in Molecules 2022, 27, 7058. https://doi.org/10.3390/molecules27207058

-“Identification, characterization and HPLC quantification of impurities in apremilast” DOI: 10.1039/C5AY01759A (Paper) Anal. Methods, 2016, 8, 1889-1897

In their own paper chiral LC separation of apremilast is also mentioned. : “Comparative Chiral Separation of Thalidomide Class of Drugs Using Polysaccharide-Type Stationary Phases with Emphasis on Elution Order and Hysteresis in Polar Organic Mode” Molecules 2022, 27, 111. https://doi.org/10.3390/molecules27010111

Please pay attention to this patent:

https://data.epo.org/publication-server/document?iDocId=6062715&iFormat=2

EUROPEAN PATENT SPECIFICATION; A METHOD OF CHIRAL RESOLUTION OF THE KEY INTERMEDIATE OF THE SYNTHESIS OF APREMILAST AND ITS USE FOR THE PREPARATION OF PURE APREMILAST, EP 3 280 701 B1

Minor comments:

line 46: please note that the CDs are not ‘dissolved’ but ‘dispersed’ in the BGE.

Author Response

The manuscript describes the separation of apremilast by CE using CDs as additives. The paper is clearly written, and the results are good, in that the authors managed to develop a CE BGE system suitable for the enantiomeric separation of apremilast. However, some validation tests are needed.

 What is the typical purity limit of the R-APR in commercial S-APR samples? Is the typical limit 0.15% or even lower? Please check the LOD of R-APR in the presence of required amounts of S-APR. In the figure caption of Figure 3, the R-APR concentration is said to be 0.1% (not 0.15% as in the text). Please give repeatability and reproducibility data (on a different capillary) for the 0.1% enantiomeric purity analysis. Please also make a robustness test.

Although the authors did not find an official upper limit for R-APR concentration, given the daily dose of APR, the typical upper limit would be 0.15%. 

Based on the reviewers' comments, the LOD and LOQ of the method were checked, and determined to be 1.0 μg/mL (corresponding to 0.05%), and 2.0 μg/mL (corresponding to 0.1%), respectively.

To evaluate the repeatability of migration times and peak areas, samples were prepared at 2.0 μg/mL (0.1%) R-APR, in the presence of the eutomer at sample concentration (2000 μg/mL). The repeatability of the method, calculated as RSD for mi-gration times were 1.00% and 0.84% for R-, and S-APR, respectively. In the case of peak area, the RSD for R-APR was 1.96% (n=6). Inter-day precision values, for migration times were RSD 1.09% and 0.86% for R-, and S-APR, respectively, while it was 2.17% for the peak area of the distomer (n=12). Repeating the analysis on a different capillary prepared from the same batch, revealed RSD values of 1.32% and 1.19% for R-, and S-APR respectively for migration times, and 2.23% for the peak area of R-APR.

These are now incorporated in the manuscript.

The Chinese patent CN107628983A https://patents.google.com/patent/CN107628983A/en seems to have a procedure for making apremilast with an enantiomer purity of more than 99.9%. Please check.

The abovementioned patent was cited at the appropriate place in the manuscript.

Please give some references to the lack of interconversion of the S-form to the L-form (this is something industrial people will be interested in). See eg. https://www.ema.europa.eu/en/documents/assessment-report/otezla-epar-public-assessment-report_en.pdf The assessment of the chiral purity in that report from 2014 was claimed to be done by HPLC.

Appropriate references were added to support the claim of lack of interconversion between the enantiomers.

Why were the samples diluted in acetonitrile? How was the type of solvent optimized? Typically, sample stock solutions for CE are done in methanol or ethanol.

During the preliminary studies, it was revealed, that the best peak shapes were obtained if the sample solution contained the least amount of organic solvent. If the sample solution contained higher concentrations of organic solvent, peak distortion or even peak splitting was observed. Thus, our approach was to dissolve the enantiomers in a suitable solvent to obtain a higher concentration stock solution, and upon further dilution, try to minimize the solvent used. ACN was chosen because it allowed us to prepare initial stock solutions of 10 mg/mL The solubility of the API was lower in the given alcohols. 

This is now incorporated in the revised version of the manuscript.

What was used as the EOF marker in the work? The EOF peaks visible in Figure 2 are probably due to ACN present in the sample. Please mark the EOF with an arrow in the electropherograms. The same holds for all the other electropherograms.

During the preliminary CD screening, 0.01% DMSO was used as the EOF marker. After Succ-b-CD was selected, no DMSO marker was employed. As the reviewer mentioned, the baseline drop, visible in the figures is due to the presence of ACN in the sample and can be considered indicative of the EOF.

This is now incorporated in the revised version of the manuscript.

Please check these recent papers, related to the topic:

 -A review paper: “New Trends in the Quality Control of Enantiomeric Drugs: Quality by Design-Compliant Development of Chiral Capillary Electrophoresis Methods” in Molecules 2022, 27, 7058. https://doi.org/10.3390/molecules27207058

 -“Identification, characterization and HPLC quantification of impurities in apremilast” DOI: 10.1039/C5AY01759A (Paper) Anal. Methods, 2016, 8, 1889-1897

 In their own paper chiral LC separation of apremilast is also mentioned. : “Comparative Chiral Separation of Thalidomide Class of Drugs Using Polysaccharide-Type Stationary Phases with Emphasis on Elution Order and Hysteresis in Polar Organic Mode” Molecules 2022, 27, 111. https://doi.org/10.3390/molecules27010111

 Please pay attention to this patent:

 https://data.epo.org/publication-server/document?iDocId=6062715&iFormat=2

 EUROPEAN PATENT SPECIFICATION; A METHOD OF CHIRAL RESOLUTION OF THE KEY INTERMEDIATE OF THE SYNTHESIS OF APREMILAST AND ITS USE FOR THE PREPARATION OF PURE APREMILAST, EP 3 280 701 B1

The abovementioned references were added to the paper, at appropriate places.

Minor comments:

 line 46: please note that the CDs are not ‘dissolved’ but ‘dispersed’ in the BGE.

Corrected.

Round 2

Reviewer 2 Report

The authors have revised the manuscript according to the comments by the reviewers.